# Impact of digital meditation on work stress and health outcomes among adults with overweight: A randomized controlled trial

**Rachel M. Radin**[1]*, **Elissa S. Epel**[1], **Ashley E. Mason**[1,2], **Julie Vaccaro**[1], **Elena Fromer**[1], **Joanna Guan**[1], **Aric A. Prather**[1]

**1** Department of Psychiatry and Behavioral Sciences, University of California, San Francisco, San Francisco, CA, United States of America, **2** Osher Center for Integrative Medicine, University of California, San Francisco, San Francisco, CA, United States of America

* Rachel.radin@ucsf.edu

**Data Availability Statement:** The data described in the manuscript, code book, and analytic code/syntax are made publicly and freely available

## Abstract

Mindfulness meditation may improve well-being at work; however, effects on food cravings and metabolic health are not well known. We tested effects of digital meditation, alone or in combination with a healthy eating program, on perceived stress, cravings, and adiposity. We randomized 161 participants with overweight and moderate stress to digital meditation ('MED,' $n = 38$), digital meditation + healthy eating ('MED+HE,' $n = 40$), active control ('HE,' $n = 41$), or waitlist control ('WL,' n = 42) for 8 weeks. Participants ($n = 145$; M(SD) BMI: 30.8 (5.4) kg/m$^2$) completed baseline and 8-week measures of stress (Perceived Stress Scale), cravings (Food Acceptance and Awareness Questionnaire) and adiposity (sagittal diameter and BMI). ANCOVAs revealed that those randomized to MED or MED+HE (vs. HE or WL) showed decreases in perceived stress ($F = 15.19$, $p < .001$, $\eta^2 = .10$) and sagittal diameter ($F = 4.59$, $p = .03$, $\eta^2 = .04$), with no differences in cravings or BMI. Those high in binge eating who received MED or MED+HE showed decreases in sagittal diameter ($p = .03$). Those with greater adherence to MED or MED+HE had greater reductions in stress, cravings, and adiposity ($ps < .05$). A brief digital mindfulness-based program is a low-cost method for reducing perceptions of stress and improving abdominal fat distribution patterns among adults with overweight and moderate stress. Future work should seek to clarify mechanisms by which such interventions contribute to improvements in health.

**Trial registration: Clinical trial registration** http://www.ClinicalTrials.gov: identifier NCT03945214.

## Introduction

Obesity remains a public health crisis [1, 2] and it is highly comorbid with work-related stress [3]. Work stress contributes to an estimated 5–8% of annual healthcare costs in the United States [4]. Epidemiological studies consistently demonstrate associations between high work stress and worse self-reported mental and physical health, including depression, anxiety, cardiovascular disease, and type 2 diabetes [5].

through Open Science Framework Repository. The
DOI Identifier is: DOI 10.17605/OSF.IO/QPG6F.

**Funding:** This work was supported by the UCSF
Healthy Campus Network; Headspace, Inc.; and the
National Center for Complementary and Integrative
Health (NCCIH) K23AT011048-01 (to RMR). The
funders had no role in study design, data collection
and analysis, decision to publish, or preparation of
the manuscript.The university's Institutional review
board (IRB) approved all aspects of this study. All
authors report no non-financial interests that could
be relevant to the submitted manuscript. Dr. Elissa
Epel is a scientific advisor to Meru Health, Inc., a
digital platform for mental health.

**Competing interests:** The authors have declared
that no competing interests exist.

Mindfulness meditation may improve well-being in workplace settings [6]. Mindfulness, in
general, aims to cultivate a non-judging awareness of experiences in the present moment and
promote adaptive self-regulation [7]. Mindfulness-based psychological interventions decrease
perceptions of stress in non-clinical populations [8], and improve psychosocial outcomes in
clinical populations with anxiety and depression [9–11]. Recent data indicate that mindful-
ness-based trainings delivered in the workplace decrease global perceptions of psychological
stress in healthy adults [12]. However, traditional in-person practice cannot be easily scaled
and disseminated, making them less cost-effective than other approaches. In the current study,
we used a commercially available digitally-delivered meditation platform.

Overeating drive patterns, such as food cravings and binge eating, may explain the links
between work stress and obesity. These eating patterns are strongly associated with obesity
[13–16] and worsened metabolic health [17] affecting up to 30% of those who seek weight-loss
treatment [18–20]. Overeating drive may uniquely predict the development of cardiovascular
and endocrine disorders, including heart disease and type 2 diabetes, even after accounting for
obesity status [17, 21]. These data support the importance of overeating drive as a behavioral
target.

Mindfulness-based approaches may be a promising avenue for targeting reductions in over-
eating drive, including food cravings, and downstream metabolic outcomes. Mindfulness-
based approaches are not diet-based, thus appealing to those with overeating drive patterns,
who may have had many unsuccessful dieting attempts. There is little data assessing whether
mindfulness approaches promote improvements in metabolic outcomes [22, 23]. It also
remains unclear for whom mindfulness-based approaches are best suited. Our prior work on a
weight loss intervention demonstrated that those with a tendency toward binge eating showed
greater improvements in a range of weight-related factors following a mindfulness interven-
tion compared to those without binge eating [24]. Thus, mindfulness-based approaches may
be a better fit for adults with obesity and overeating drive, in comparison to standard behav-
ioral weight loss interventions.

Mindfulness training delivered via a self-guided smartphone app may offer a convenient
alternative to in-person treatment, though research on their efficacy is limited [25, 26]. Three
small studies using smartphone apps to deliver mindfulness interventions to healthy adults
found benefits comparable to traditional delivery methods on subjective well-being, depressive
symptoms, and compassion [27–29]. App-based interventions also offer the benefit of stan-
dardization of instruction across participants, as well as the ability for participants to control
where and when they access the intervention, and objective measures of adherence, rather
than self-report. Digital mindfulness interventions demonstrate significant reductions in per-
ceived stress and increases in subjective mindfulness, compared to a wait list condition,
among non-clinical populations [30]. A recent meta-analysis of digital occupational mental
health interventions [31], which included mindfulness-based programs, found small, positive
effects on psychological well-being and work effectiveness.

Treatment adherence to digitally-based mindfulness interventions is an understudied, yet
probable moderator of treatment effects. In a recent 8-week pilot, Carolan and colleagues [32]
found greater treatment engagement in digital programs incorporating a discussion group. It
is also unknown whether digitally-based mindfulness interventions improve overeating drive
or metabolic health. A recent meta-analysis [33] of in-person work-based mindfulness medita-
tion programs found them generally effective in lowering cortisol production, heart rate, and
sympathetic activity. Previous work using in-person mindfulness has shown that mindful eat-
ing training reduces abdominal fat without reducing overall body mass index [34, 35].

*The Current Study*: To test the effects of digital meditation on stress, cravings, and abdomi-
nal adiposity, we tested 4 conditions including an active control group with information about

healthy eating, and a no treatment wait list control. The healthy eating program, which we considered to be an "active control" condition, utilized mindfulness-based and motivational approaches to improve eating behaviors. We aimed to test whether digital meditation could out-perform an active control condition that was matched for time and attention and other non-specific intervention effects [36]. We were interested in whether the adjunctive treatment offered from the healthy eating program would further improve outcomes compared to digital mindfulness alone.

We aimed to examine the effects of treatment randomization on global perceptions of psychological distress [37] and overeating drive [38]. Secondarily, we examined treatment effects on body mass index (BMI) and sagittal diameter. Finally, we examined the influence of treatment adherence (total minutes participants engaged in meditation on the app) on treatment outcomes. We hypothesized that mindfulness (in either form) would out-perform either control condition with respect to improvements in primary and secondary outcomes, and that treatment adherence would moderate these effects. We also anticipated that the combination of digital mindfulness + healthy eating (vs. digital mindfulness alone) would promote the greatest improvements. We also endeavored to examine the potential moderating role of binge eating presence. We hypothesized that those with binge eating would derive the greatest benefit from a mindfulness-based digital intervention, whereas those without binge eating would show no differences in outcomes across interventions.

## Materials and methods

### Study overview

We aimed to test the effects of a digital meditation intervention vs. an active or wait list control, on subjective measures of perceived stress, food cravings, and adiposity in a sample of employees at a large university with overweight and obesity who reported mild to moderate stress (NCT03945214). We randomized participants to 8-weeks of a digital meditation intervention (using the commercially available application, Headspace), a healthy eating intervention (active control), a digital meditation + healthy eating intervention, or a waitlist control condition. We asked all participants to complete questionnaires and anthropomorphic measurements at an in-person clinic visit at baseline and week 8. Adherence to the digital meditation intervention was tracked remotely by Headspace.

### Participants

Eligible participants were ≥18 years old, employed at a large academic medical center, had a BMI equal to or greater than 25 kg/m$^2$, reported mild to moderate levels of stress in the previous month (as determined by a Perceived Stress Scale score of 15 or higher), and had daily access to a smartphone or computer. Exclusion criteria included being an experienced meditator (defined as 3 times per week for 10 minutes or more). We obtained written informed consent from all study participants. We aimed to enroll up to 150 participants. Our prior study [39] detected effects in a sample of <250 participants. We therefore expected that our sample size of 150 would be well-powered to detect improvements in our self-report measures in response to our treatment intervention. The university's Institutional review board (IRB) approved all aspects of this study. Participants did not receive monetary compensation; however they received a one-year subscription to Headspace (value of $150), and were entered into a raffle drawing to win a 2-night expenses paid vacation in the local Bay Area.

## Study design

Participants completed baseline assessment procedures, including measures of body composition and self-report assessments. Study personnel then randomly assigned participants to one of four possible conditions, using factorial assignment, on Qualtrics: (1) Meditation only, (2) Healthy Eating only, (3) Meditation + Healthy Eating, or (4) Waitlist control. The sequence of assignments was generated ahead of time with a computer script by a statistician who was not involved in running the study. Study personnel were not able to access the file containing the sequence of assignments or to see the next condition in the sequence until the moment they randomized the participant. Both participants and study staff were unblinded to the assignment after allocation. We re-assessed participants on all measures collected at baseline (body composition, self-report assessments) again at 8 weeks from randomization.

## Interventions

**Meditation group ('MED').** We provided participants with access to digitally-based meditation program (Headspace app- Basics + 'Letting go of stress' packs) and asked them to engage with the app for at least 10 minutes a day for 8 weeks. We contacted participants who completed less than one meditation in the previous 10 to 17 days via phone, in order to re-engage them with the program. Participants were expected to meditate 5 days per week over the course of 8 weeks.

**Healthy eating group ('HE').** Within week 1, we provided participants with an in-person 50-minute counseling session with a trained health counselor geared towards developing goals to improve eating behavior, along with three 10-minute booster phone calls at weeks 1, 4, and 8. The counseling session incorporated a motivational interviewing framework to assess areas of concern around eating behavior and to establish specific and achievable eating-related goals. We also asked participants to engage with a digitally-based mindful eating program once per week for 8 weeks, and sent text message reminders 3 times per week to increase accountability towards eating-related goals. The digitally-based mindful eating program was created specifically for this study by the research team, and was primarily a secured website that included information on mindful eating, and audio tools for mindful eating practice (~3–5 minute practices). This password-secured website contained up to six different brief audio exercises using mindful eating and urge-surfing strategies. The audio exercises were scripted and recorded by the study's first author and adapted from a number of mindful eating resources including the mindfulness-based eating awareness training (MB-EAT) curriculum [45]. We instructed participants to access these audios during high vulnerability times for compulsive eating. For example, for those who identify cravings as a potent trigger for problematic consumption, participants could access a brief urge-surfing exercise to learn how to 'ride out' a craving. Participants had a total of approximately 1.5–2 hours of contact with a counselor, and were expected to engage with the online resources 1 day per week over the course of 8 weeks. The program is adapted from several sources, including motivational interviewing for binge eating, weight management, and sugar-sweetened beverage intake (from our recently completed trial), and mindfulness-based eating awareness training.

**Meditation + healthy eating group ('MED+HE').** We provided participants with access to digitally-based meditation program (as described above under 'MED') in addition to the 'HE' program (as described above under 'HE'). Participants were expected to meditate 5 days per week over the course of 8 weeks and they had a total of approximately 1.5–2 hours of contact with a counselor, and were expected to engage with the online resources 1 day per week over the course of 8 weeks.

**Waitlist control condition ('WL').** We instructed participants to continue their normal activities and not add any meditation during the study period. We did not provide Headspace

access codes to WL or HE participants until after they completed a 2-month follow-up questionnaire. Participants had no contact with a study counselor over the course of the 8 week intervention period.

## Measures

**Primary outcome measures.** *Perceived stress.* The Perceived Stress Scale (PSS; [37] is a 10-item self-report questionnaire that measures a persons' evaluation of the life stress they have experienced over the previous month, and has been extensively validated. The PSS has a total score scale range of 0 to 40, with higher values indicating more perceived stress. The PSS has demonstrated adequate reliability and validity among similar populations [37]. Among our study sample, scale reliability was high (α = .87)

**Tolerance for food cravings.** The Food Acceptance and Awareness Questionnaire (FAAQ) measures acceptance of urges and cravings to eat or the extent to which individuals might try to control or change these thoughts [38]. The FAAQ is made up of 10 items, each rated on a 6-point Likert scale (1 = *very seldom true* to 6 = *always true*). It has a total score scale range of 10 to 60, with higher scores indicating greater acceptance of motivations to eat and greater tolerance for food cravings. The FAAQ has demonstrated sound psychometric properties [38]. Among our study sample, scale reliability was high (α = .80).

**Secondary outcome measures.** *BMI.* We calculated body mass index (BMI) as weight in kilograms divided by the square of height in meters (kg/m$^2$). Weight was measured twice using a digital scale, and height was measured using a stadiometer.

*Sagittal diameter.* We measured body fat distribution using an abdominal caliper placed just above the umbilicus, measuring the distance from the small of the back to the upper abdomen. Measurements were taken, using the two closest measurements that were within 0.5 cm, and recorded to the nearest 0.1 cm.

*Binge presence.* We used the Questionnaire on Eating and Weight Patterns –5 (QEWP-5) to determine the presence of binge eating. The QEWP-5 [40] is a 24-item questionnaire that assesses frequency of reported binge eating and loss of control eating episodes, which has been shown to have reasonable agreement with interview-based measures such as the Eating Disorder Examination [40]. Binge presence was defined by the endorsement of the following: 1-*During the last 3 months, did you ever eat, in a short period of time- for example, a two hour period- what most people would think was an unusually large amount of food?; 2- During the times when you ate an unusually large amount of food, did you often feel you could not stop eating or control what or how much you were eating?*

**Treatment adherence.** Adherence to either meditation program (MED or MED+HE) was calculated by summing the total number of minutes spent meditating via Headspace over 8 weeks. The research team had access to individual user data via Headspace, in order to make these calculations. We also assessed meditation frequency with the following questions: "How often did you practice sitting meditation (for 10 min or more) in the past 8 weeks?" Participants selected of the following options: never, less than once a week, 1–3 times per month, 1–2 times per week, 3–4 times per week, or every day. We used this information to ensure that those in the control conditions (active and wait list control) abstained from meditation practice throughout the intervention period.

## Statistical analysis

**Data preparation.** We used SPSS (Version 27.0. Armonk, NY: IBM Corp.) for all variable preparation and statistical analysis. We computed summary statistics to evaluate the distributions of each study variable (i.e., PSS, FAAQ, BMI, sagittal diameter, binge presence, treatment

adherence) and assess potential outliers. We did not find any outliers with regard to primary or secondary outcome variables (defined as $> \pm 3$ standard deviations of the mean).

**Treatment effect on outcome variables.** In a series of Analysis of Covariance (ANCOVA) models, we compared treatment groups (IV: MED vs. MED+HE vs. HE vs. WL) on each 8-week outcome variable (DV: Treatment adherence, PSS, FAAQ, BMI, Sagittal Diameter), adjusting for baseline value of each corresponding measure (covariate). If the main ANCOVA model was significant, we used post-hoc (least square differences) tests to explore group differences. In sub-analyses, we ran an identical series of ANCOVA models, where we collapsed treatment groups (IV) into 'meditation' (MED or MED+HE) vs. 'no meditation' (HE or WL).

**Moderation analyses.** We ran a series of ANCOVA models adding an interaction term between treatment group and total meditation minutes (treatment adherence) and examined the simple slopes of the interaction term. We also ran a series of linear regressions to explore whether baseline binge presence (treated as a dichotomous variable of binge vs. no binge presence) moderated the effect of treatment group on primary and secondary outcome variables. We created an interaction term (between binge presence at baseline X intervention) as our independent variable. In all analyses, we considered $p \leq .05$ to be statistically significant (using two-tailed tests).

## Results

### Participant recruitment and retention

We enrolled 161 participants, who we randomized to: MED ($n = 38$), MED+HE ($n = 40$), HE ($n = 41$), or WL ($n = 42$). At 8 weeks, 145 participants completed follow-up surveys and 128 participants completed an in-person follow-up visit (see **Fig 1** for CONSORT diagram).

### Participant characteristics

Participants had a mean BMI of 30.78 kg/m$^2$ (40% with obesity vs. 60% with overweight). The majority (40%) identified as White, and reported a four-year college or graduate degree (85%). We classified the majority of participants as administrative staff (30%), researchers (19%) mid-level managers (16%) or medical staff (15%). By study design, participants endorsed a mean PSS score indicative of moderate stress (37) and the majority (>95%) reported meditating less than once a week. Approximately 39% endorsed binge eating presence (objectively large amount of food + loss of control; **Tables 1 and 2** for demographic and health characteristics of the sample, respectively).

### Treatment adherence

Participants randomized to MED or MED+HE ($n = 78$) engaged with the Headspace app an average of $4.15 \pm 4.22$ minutes per day with no differences between meditation groups ($t = 1.50$, $p = .14$). Approximately 10% ($n = 8$) were adherent to instructions to meditate $\geq 10$ minutes per day over the course of the 8 week program. Participants randomized MED or MED+HE (vs. HE or WL) reported a greater frequency of meditation at 8 weeks, after accounting for baseline frequency ($F = 78.51$, $p < .001$). The majority of those in MED (83%) or MED+HE (72%) reported meditating up to two times per week at 8 weeks (compared to 9% of those in HE and 3% of those in WL), suggesting that both mindfulness groups were adherent to treatment (i.e., engaging in mindfulness).

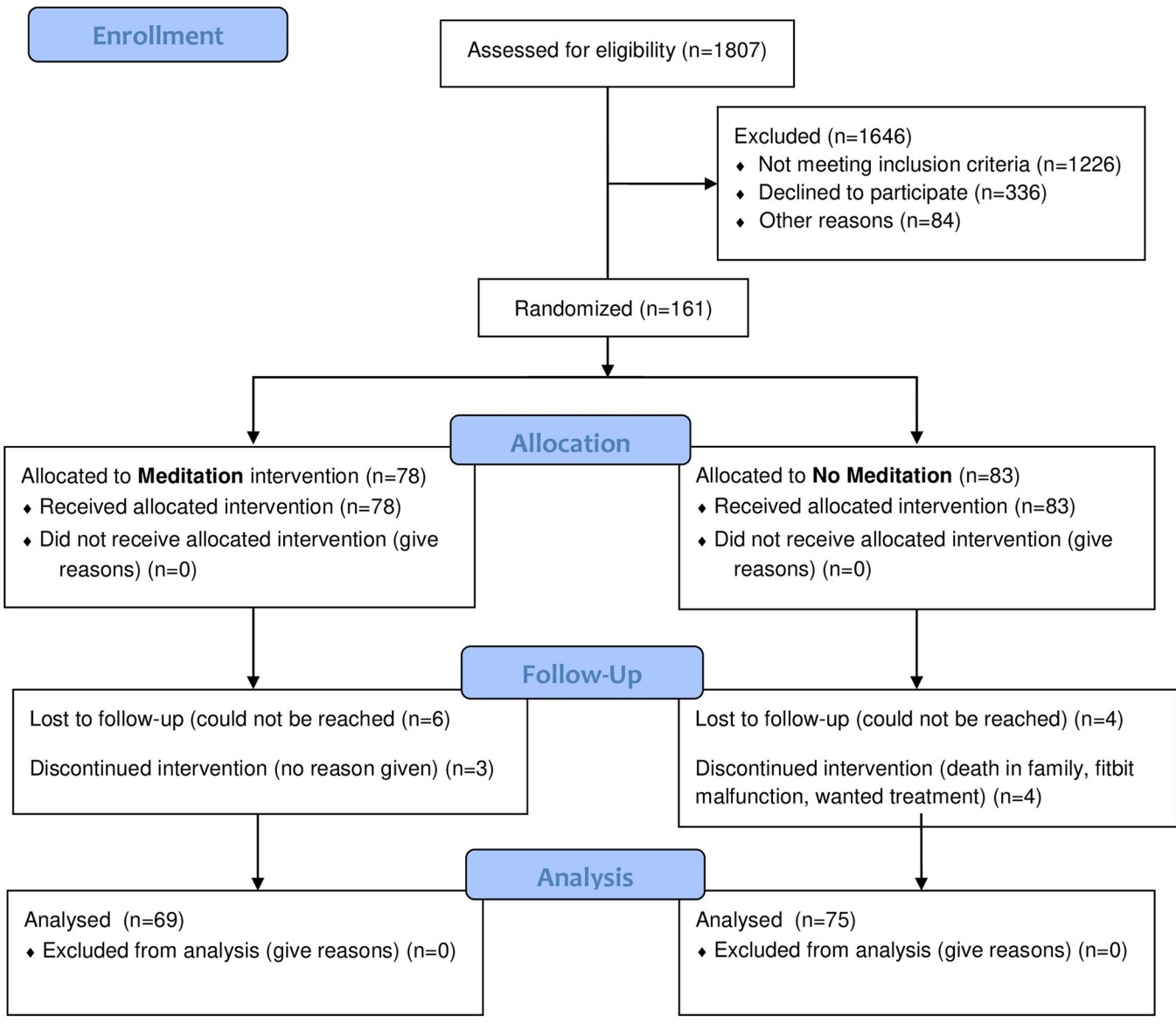

**Fig 1. CONSORT flow diagram.**

## Primary outcomes

**Perceived stress.** Among all 4 groups, there was a treatment effect ($F(3,139) = 5.91$, $p = .001$, $\eta^2 = .11$), such that those in MED (mean change: -5.97, SE = 0.94, 95% CI: -7.84, -4.11) or MED+HE (mean change: -4.97, SE = 0.99, 95% CI: -6.92, -3.02) showed the greatest decreases in PSS score (with no differences between MED vs. MED+HE, $p = .30$), compared to those in HE (mean change: -2.00, SE = 0.93, 95% CI: -3.84, -0.16) or WL (mean change: -1.66, SE = 0.92, 95% CI: -3.48, 0.16); with no differences between HE vs. WL, $p = .80$; **Fig 2**). In sub-analyses, those randomized to either 'meditation' (i.e., MED or MED+HE) group showed greater decreases in PSS score (26% reduction) vs. those in either 'no meditation (i.e., HE or WL)' group (8% reduction; $F(1,142 = 15.19$, $p < .001$, $\eta^2 = .10$). Findings were identical when using non-parametric tests (Kruskal-Wallis), given the ordinal nature of the PSS scoring.

**Table 1. Baseline demographics.**

| Variable | n | Total Sample | n | Meditation | n | Meditation + Healthy Eating | n | Healthy Eating | n | Waiting list Control |
|---|---|---|---|---|---|---|---|---|---|---|
| Demographics | | | | | | | | | | |
| Age (years) (M±SD) | 161 | 37.92 ± 11.18 | 38 | 38.63 ± 11.01 | 40 | 35.40 ± 8.03 | 41 | 39.29 ± 12.43 | 42 | 38.33 ± 12.55 |
| Sex (% female) | 161 | 72 | 38 | 71.05% | 40 | 67.50% | 41 | 82.93% | 42 | 66.67% |
| Race/Ethnicity (%): | 161 | | 38 | | 40 | | 41 | | 42 | |
| White | | 39.8 | 16 | 42.11% | 14 | 35.00% | 14 | 34.15% | 20 | 47.62% |
| Black or African American | | 9.3 | 3 | 7.89% | 5 | 12.50% | 4 | 9.76% | 3 | 7.14% |
| Hispanic or Latino | | 12.4 | 7 | 18.42% | 6 | 15.00% | 5 | 12.20% | 2 | 4.76% |
| Asian/Pacific Islander | | 21.1 | 6 | 15.79% | 8 | 20.00% | 7 | 17.07% | 13 | 30.95% |
| Multiple races | | 13.7 | 4 | 10.53% | 7 | 17.50% | 7 | 17.07% | 4 | 9.52% |
| Other | | 3.7 | 2 | 5.26% | 0 | 0.00% | 4 | 9.76% | 0 | 0.00% |
| Education (y) (%): | 161 | | 38 | | 40 | | 41 | | 42 | |
| Less than 4 year degree | | 11.8 | 6 | 15.78% | 5 | 12.50% | 4 | 9.76% | 4 | 9.52% |
| 4 year degree | | 39.1 | 12 | 31.58% | 16 | 40.00% | 19 | 46.34% | 16 | 38.10% |
| Professional degree | | 32.3 | 11 | 28.95% | 12 | 30.00% | 14 | 34.15% | 15 | 35.71% |
| Doctorate | | 13.7 | 7 | 18.42% | 6 | 15.00% | 4 | 9.76% | 5 | 11.90% |
| No response | | 3.1 | 2 | 5.26% | 1 | 2.50% | 0 | 0.00% | 2 | 4.76% |
| Annual household income (%): | 161 | | 38 | | 40 | | 41 | | 42 | |
| Less than $35,000 | | 4.4 | 3 | 7.89% | 1 | 2.50% | 2 | 4.88% | 1 | 2.38% |
| $35,000 to less than $50,000 | | 5.0 | 1 | 2.63% | 1 | 2.50% | 3 | 7.32% | 3 | 7.14% |
| $50,000 to less than $75,000 | | 20.5 | 9 | 23.68% | 10 | 25.00% | 4 | 9.76% | 10 | 23.81% |
| $75,000 to less than $100,000 | | 18.0 | 5 | 13.16% | 8 | 20.00% | 11 | 26.83% | 5 | 11.90% |
| $100,000 to less than $150,000 | | 21.1 | 8 | 21.05% | 10 | 25.00% | 8 | 19.51% | 8 | 19.05% |
| $150,000 to less than $200,000 | | 14.9 | 6 | 15.79% | 3 | 7.50% | 7 | 17.07% | 8 | 19.05% |
| $200,000 or more | | 11.8 | 4 | 10.53% | 6 | 15.00% | 5 | 12.20% | 4 | 9.52% |
| Prefer not to answer/no response | | 4.3 | 2 | 5.26% | 1 | 2.50% | 1 | 2.44% | 3 | 7.14% |

Frequency of meditation moderated the effect of treatment on changes in PSS (interaction term, $F = 4.74$, $p = .03$), such that greater treatment adherence in meditation was associated with greater decreases in PSS score at 8 weeks ($r = -.27$, $p = .03$).

**Tolerance for food cravings.** Among all 4 groups, we found no treatment effect ($F_{(3,132)} = 0.58$, $p = .63$, $\eta^2 = .01$). Comparing the estimated marginal means showed a pattern (while not significant) that those in HE showed the greatest increases in FAAQ (mean change: +1.80, SE = 1.37, 95% CI: -0.91, 4.51), followed by those in WL (mean change: +0.81, SE = 1.33, 95% CI: -1.83, 3.45), MED (mean change: +0.26, SE = 1.37, 95% CI: -2.46, 2.97) and MED+HE (mean change: -0.83, SE = 1.51, 95% CI: -3.81, 2.15; Fig 3). In sub-analyses, those randomized to 'meditation' vs. 'no meditation' did not differ; Both groups showed similar changes (meditation: 0.10% reduction; vs no meditation: 4.3% increase; $F_{(1,134} = 1.21$, $p = .27$, $\eta^2 = .01$). Findings were identical when using non-parametric tests (Kruskal-Wallis), given the ordinal nature of the FAAQ scoring. Frequency of meditation did not moderate the effect of treatment on changes in FAAQ ($p > .10$). However, we observed a main effect of meditation frequency on FAAQ at 8-weeks ($F = 5.31$, $p = .02$), irrespective of treatment randomization. Treatment adherence was associated with higher FAAQ scores at 8-weeks ($r = .27$, $p = .03$), although it was not associated with changes in FAAQ score at 8 weeks ($r = .20$, $p = .12$).

**Table 2. Baseline health characteristics.**

| Variable | | Total Sample | | Meditation | | Meditation + Healthy Eating | | Healthy Eating | | Waiting list Control |
|---|---|---|---|---|---|---|---|---|---|---|
| | *n* | | *n* | | *n* | | *n* | | *n* | |
| Physiological Characteristics (M±SD): | | | | | | | | | | |
| BMI (kg/m$^2$) | 161 | 30.78 ± 5.43 | 38 | 30.10 ± 4.49 | 40 | 31.17 ± 6.39 | 41 | 30.98 ± 5.07 | 42 | 30.80 ± 5.69 |
| Sagittal Diameter (cm) | 161 | 25.73 ± 5.14 | 38 | 24.85 ± 5.00 | 40 | 25.88 ± 5.80 | 41 | 26.01 ± 5.08 | 42 | 26.11 ± 4.75 |
| Stress, Psychological Measures (M±SD): | | | | | | | | | | |
| Perceived Stress Scale | 160 | 21.88 ± 4.84 | 38 | 22.00 ± 5.59 | 40 | 22.38 ± 4.68 | 41 | 21.68 ± 4.36 | 41 | 21.46 ± 4.86 |
| Meditation frequency (%) | 161 | | 38 | | 40 | | 41 | | 42 | |
| Never | | 78.9 | | 68.4 | | 85 | | 80.5 | | 81 |
| Less than once a month | | 9.3 | | 18.4 | | 5 | | 7.4 | | 7.1 |
| 1–3 times a month | | 7.5 | | 7.9 | | 5 | | 7.3 | | 9.5 |
| 1–2 times a week | | 4.3 | | 5.3 | | 5 | | 4.9 | | 2.4 |
| Eating Measures (M±SD): | | | | | | | | | | |
| Food Acceptance and Action Questionnaire (FAAQ) | 155 | 29.20 ± 6.49 | 37 | 29.41 ± 6.56 | 38 | 29.53 ± 6.69 | 41 | 28.78 ± 6.37 | 39 | 29.13 ± 6.56 |
| Binge eating presence (%) (QEWP)[a] | 160 | 38.5 | 38 | 34.21% | 40 | 37.50% | 41 | 46.34% | 42 | 35.71% |
| # Binge episodes per week (QEWP) | 160 | 0.91 ± 1.40 | 38 | 0.84 ± 1.35 | 39 | 0.97 ± 1.55 | 41 | 1.12 ± 1.47 | 42 | 0.71 ± 1.24 |

[a] QEWP = Questionnaire on Eating and Weight Patterns [38]. Binge presence was defined by the endorsement of the following: 1- *During the last 3 months, did you ever eat, in a short period of time- for example, a two hour period- what most people would think was an unusually large amount of food*?; 2- *During the times when you ate an unusually large amount of food, did you often feel you could not stop eating or control what or how much you were eating*?

## Secondary outcomes

**Sagittal diameter.** Among all 4 treatment groups, we found no treatment effect ($F(3,124)$ = 1.69, $p$ = .18 = 7; $\eta^2$ = .04). Comparing the estimated marginal means showed a pattern (while not significant) that those in MED+HE (mean change: -0.25, SE = 0.24, 95% CI: -0.72,

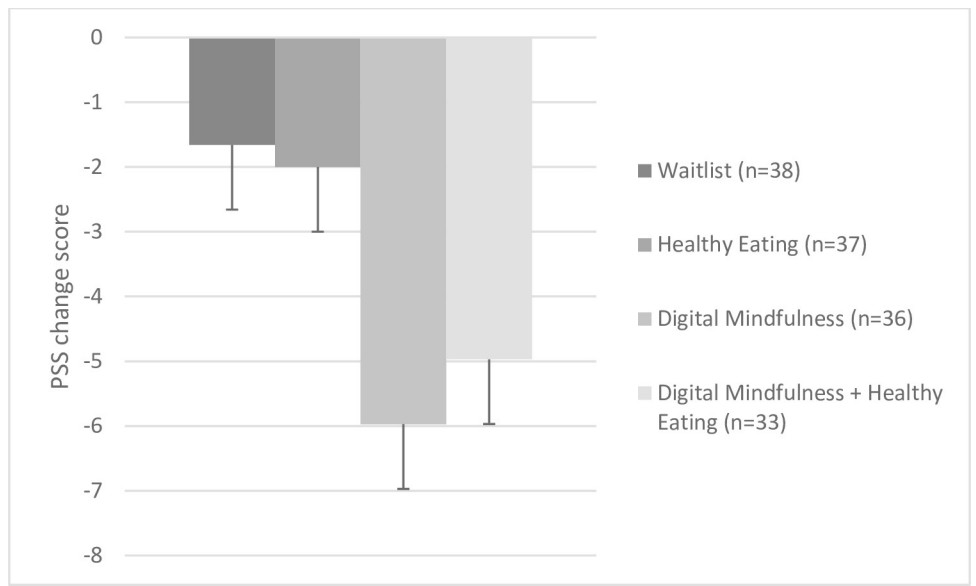

**Fig 2. Effect of treatment randomization on perceived stress (PSS) at 8 weeks, accounting for baseline values.**

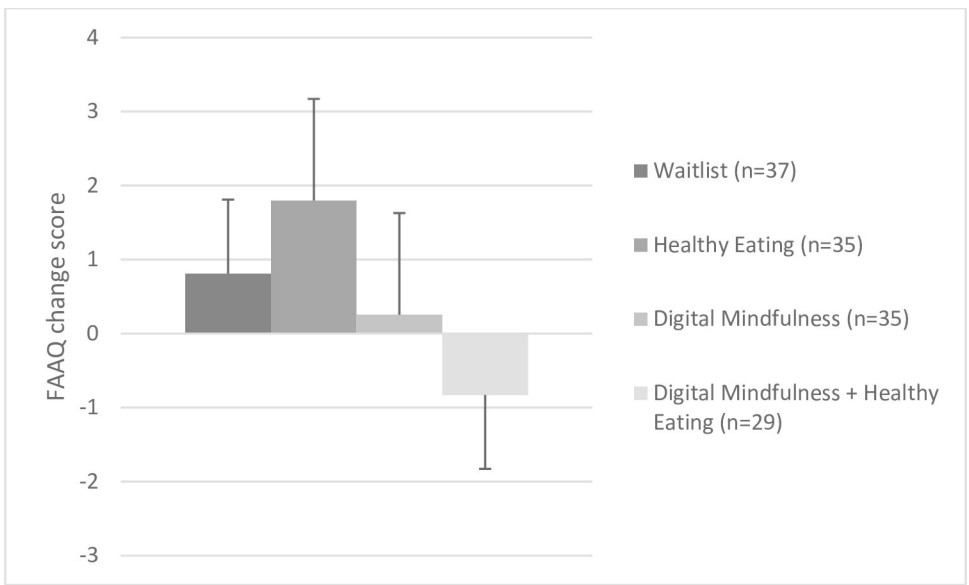

**Fig 3. Effect of treatment randomization on Tolerance for Food Cravings (FAAQ) at 8 weeks, accounting for baseline values.**

0.22) and MED (mean change: -0.12; SE = 0.23, 95% CI: -0.57, 0.33) showed decreases in sagittal diameter, whereas those in HE (mean change:+0.41, SE = 0.23, 95% CI: -0.05, 0.86) and WL (mean change: +0.21, SE = 0.23, 95% CI: -0.24, 0.66) showed slight increases. In sub-analyses, those randomized to either 'meditation' group showed greater decreases in sagittal diameter (-0.19 cm; 1% reduction) vs. those in either 'no meditation' group (+0.31 cm; 1% increase; $F_{(1,126)}$ = 4.59, $p$ = .03; $\eta^2$ = .04, **Fig 4**). Frequency of meditation did not moderate the effect of treatment randomization on changes in sagittal diameter. However, we observed a main effect

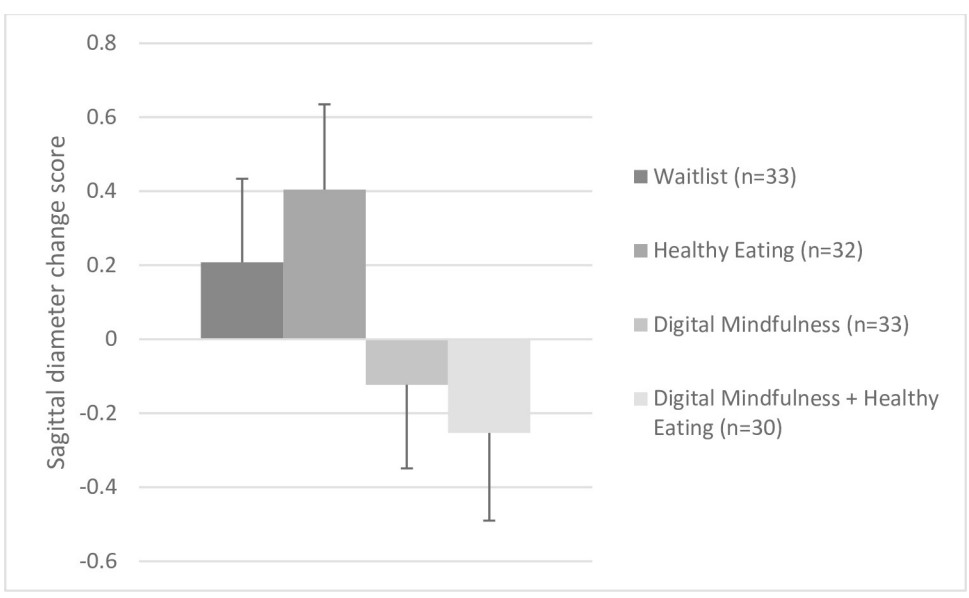

**Fig 4. Effect of treatment randomization on sagittal diameter at 8 weeks, accounting for baseline values.**

of meditation frequency on sagittal diameter at 8-weeks ($F = 15.21$, $p < .001$), irrespective of treatment randomization. Treatment adherence was associated with greater decreases in sagittal diameter at 8 weeks ($r = -.45$, $p < .001$).

**BMI.**   Among all 4 treatment groups, we found no treatment effect ($F(3,124) = 1.61$, $p = .19$, $\eta^2 = .04$) Comparing the estimated marginal means showed a pattern (while not significant) that those in MED+HE (mean change: -.66, SE = 0.29, 95% CI: -1.25, -0.08) showed slight decreases in BMI, whereas those in HE (mean change: +0.04, SE = 0.28, 95% CI: -0.52, 0.60), WL (mean change: +0.06, SE = 0.28, 95% CI: -0.50, 0.61) and MED (mean change: +0.11, SE = 0.28, 95% CI: -0.44, 0.67) showed slight increases. In sub-analyses, those randomized to 'meditation' vs. 'no meditation' did not differ; Both groups showed similar changes (meditation: -0.26 kg/m$^2$; 1% reduction; vs no meditation: +0.05; 1% reduction; $F(1,126) = 1.13$, $p = .29$; $\eta^2 = .01$). Frequency of meditation did not moderate the effect of treatment randomization on changes in BMI. Treatment adherence was not associated with changes in BMI at 8-weeks ($r = -.03$, $p = .83$).

**Moderation by baseline binge eating status.**   We did not observe a main effect of treatment randomization on binge presence at 8 weeks, (chi$^2$ = 0.78, $p = .46$). We did not find evidence for a moderating effect of baseline binge presence on our primary outcome variables (PSS, FAAQ, *ps for interaction terms*>.50). However, baseline binge presence moderated the effect of treatment randomization on changes in sagittal diameter at 8 weeks ($F(1,123) = 4.95$, $p = .03$, $\eta^2 = .04$). Examining the simple slopes of this interaction term showed that the association between treatment randomization and sagittal diameter was stronger among those with binge presence but not among those without binge presence. Participants with baseline binge presence showed greater decreases in sagittal diameter if randomized to the meditation (vs. no meditation) group, whereas participants without binge presence did not differ in sagittal diameter changes based on treatment randomization (**Fig 5**). We observed a similar interaction effect on changes in BMI, although this effect approached statistical significance ($F(1,123) = 3.09$, $p = .08$, $\eta^2 = .03$), such that those who reported binge presence tended to derive the greatest benefit when randomized to meditation vs. no meditation.

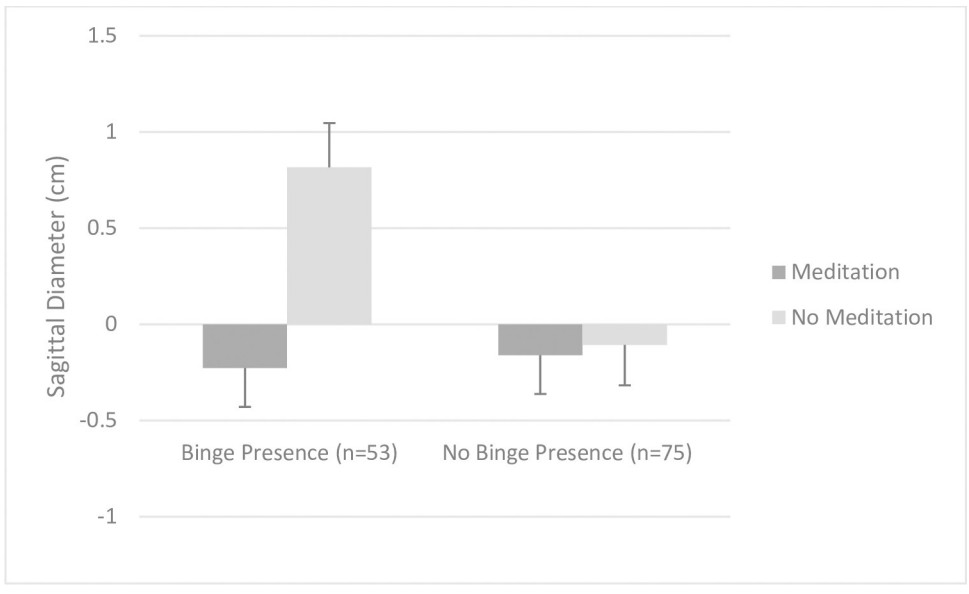

**Fig 5.  Associations between treatment randomization and changes in sagittal diameter at 8 weeks among those with (*n* = 53) vs. without (*n* = 75) baseline binge presence.**

## Discussion

Participants with overweight and moderate stress, who received either one of the digitally-based mindfulness programs showed expected reductions in perceived stress, thus confirming prior findings [30, 31]. We also found a small, but significant treatment effect on reductions in sagittal diameter. Contrary to our hypothesis, there was no treatment effect on food cravings or BMI. In an exploratory analysis, we found that meditators who also reported binge eating significantly reduced sagittal diameter.

We found preliminary evidence for a moderating effect of treatment adherence on reductions in perceived stress. Furthermore, meditation frequency was positively associated with greater tolerance for food cravings and decreases in sagittal diameter. It is plausible that treatment adherence, measured by meditation frequency using the Headspace app, accounts for treatment effects in a dose-like fashion, and suggests a mechanistic pathway, promoting reductions in stress, food cravings, and abdominal fat.

Few digitally-based mindfulness interventions have examined treatment effects on weight and metabolic outcomes [33]. We found a small but significant treatment effect on reductions in sagittal diameter, despite no reductions in BMI. In-person mindfulness interventions have improved some physiological outcomes, including blood pressure, glucose, and abdominal fat [34, 35, 41, 42] despite no changes in BMI. To our knowledge, this is this first digitally-based mindfulness intervention to observe such an effect on abdominal fat distribution. Given the main effect of treatment adherence on reductions in sagittal diameter, it is plausible that this effect is mediated by stress-related pathways, including reductions in cortisol. We found that reduction in perceived stress were associated with reductions in sagittal diameter and increases in awareness of food cravings ($ps \leq .05$). It is plausible that participants who received digital mindfulness may make healthier eating choices (e.g., increased mindfulness around satiety/hunger) which may contribute to downstream metabolic improvements. However, we were not adequately powered to test such a mechanistic pathway. This finding may also point to the importance of measuring abdominal fat distribution, in addition to BMI, in mindfulness-based digital trials.

Contrary to our hypothesis, we did not find a treatment effect on food cravings. However, the mindfulness groups reduced in binge eating (although this finding approached statistical significance). Thus, we were unable to replicate known effects of in-person mindfulness-based and mindful eating-based interventions on reductions in dysregulated eating [43–46]. Further, the addition of a healthy eating program did not add a beneficial effect to our primary or secondary outcomes. Both digital mindfulness groups (alone or with healthy eating) performed equally well with regard to reductions in perceived stress and sagittal diameter.

It should be noted that the healthy eating (active control) program was newly developed, and in need of further refining following feasibility and acceptability testing. Participants randomized to healthy eating showed good adherence (only 4% declined participation following the initial counseling session). The majority (73%) of participants rated the program as 'good' to 'excellent', and 91% of completers would recommend the program. Thus, while the healthy eating program, as packaged, did not reduce our measures of food craving, the feasibility data provided the necessary preliminary evidence for future refinement and testing. Qualitative data point to the potential added value of face-to-face counseling to establish health-related eating goals. It is plausible that participants first need to learn general mindfulness skills before showing eating-related improvements.

Finally, while exploratory in nature, we replicated our prior findings with regard to treatment matching [24], such that participants with baseline binge presence showed the greatest decreases in sagittal diameter in the meditation (vs. no-meditation) group. These findings

suggest that mindfulness may be a better fit for adults with overweight <u>and</u> overeating drive, in comparison to treatment as usual. We did not actively recruit participants high in binge eating, although nearly 40% endorsed engaging in some level of binge eating. Future RCTs should specifically seek to recruit adults with both overweight and binge eating, to fully examine whether mindfulness-based digital approaches contribute to greater improvements in psychological and metabolic health among this high-risk group.

This study had several strengths. We were able to deliver a primarily self-guided, scalable treatment for meditation to adults who experienced both perceived stress and overweight. We observed generally good adherence to our digital intervention, with only 11% being lost to follow-up. We had the added benefit of being able to compare our treatment (mindfulness) to what we considered to be an active control (healthy eating) matched for time and attention. However, our study was likely limited by a sample size that may have been too small to detect modest interaction effects. We were unable to truly ascertain whether participants in either control condition were accessing mindfulness programs or apps during the 8 week intervention period. Further, our measures of dysregulated eating may not fully reflect non-homeostatic eating behavior (vs. a semi-structured interview measure of eating pathology), and the scoring metrics for the FAAQ (a 6-point Likert scale ranging from 1 to 6) without the option for including negative (e.g., -1, -2) response may not yield particularly meaningful arithmetic means. Finally, our sample of participants were highly educated and primarily White. Thus, our findings may not fully generalize to the US population of adults with overweight.

## Conclusions

A brief digital mindfulness-based intervention is a low-cost method to reduce perceived measures of stress and may have the potential to reduce abdominal fat distribution among adults with overweight and moderate stress. These findings add to the existing literature documenting salutary effects of mindfulness on reports of well-being and extends it to digital-based mindfulness interventions. Future work should seek to clarify mechanisms by which digitally-based mindfulness interventions may contribute to improvements in psychological and physiological health.

## Supporting information

**S1 Checklist. CONSORT 2010 checklist of information to include when reporting a randomised trial**∗.
(DOC)

**S1 File.**
(DOCX)

**S2 File.**
(PDF)

## Acknowledgments

We gratefully acknowledge our participants for donating their time to this study.

## Author Contributions

**Conceptualization:** Rachel M. Radin, Elissa S. Epel, Aric A. Prather.

**Data curation:** Rachel M. Radin.

**Formal analysis:** Rachel M. Radin.

**Funding acquisition:** Rachel M. Radin, Elissa S. Epel, Aric A. Prather.

**Investigation:** Rachel M. Radin, Elissa S. Epel, Elena Fromer, Aric A. Prather.

**Methodology:** Rachel M. Radin, Aric A. Prather.

**Project administration:** Julie Vaccaro, Elena Fromer, Joanna Guan.

**Supervision:** Aric A. Prather.

**Validation:** Julie Vaccaro.

**Writing – original draft:** Rachel M. Radin, Elissa S. Epel, Aric A. Prather.

**Writing – review & editing:** Rachel M. Radin, Ashley E. Mason, Julie Vaccaro, Elena Fromer, Joanna Guan.

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
