## [Decision Letter · Decision Letter 0]

20 Jul 2022

PONE-D-21-38719Impact of digital meditation on work stress and health outcomes among adults with overweight: A randomized, controlled trialPLOS ONE

Dear Dr. Radin,

Thank you for submitting your manuscript to PLOS ONE. After careful consideration, we feel that it has merit but does not fully meet PLOS ONE’s publication criteria as it currently stands. Therefore, we invite you to submit a revised version of the manuscript that addresses the points raised during the review process.

The manuscript has been evaluated by two reviewers, including one statistical review. Both reviewers mention some minor points which should be addressed before resubmission of the manuscript. Could you please revise the manuscript to address all their concerns, including provision of the CONSORT checklist?

We look forward to receiving your revised manuscript.

Kind regards,

Thomas Tischer

Staff Editor

PLOS ONE

Journal Requirements:

“This work was supported by the UCSF Healthy Campus Network; Headspace, Inc.; and the National Center for Complementary and Integrative Health (NCCIH) K23AT011048-01 (to RMR). The funders had no role in study design, data collection and analysis, decision to publish, or preparation of the manuscript.The university’s Institutional review board (IRB) approved all aspects of this study.”

Reviewers' comments:

Reviewer's Responses to Questions

**Comments to the Author**

1. Is the manuscript technically sound, and do the data support the conclusions?

Reviewer #1: Yes

Reviewer #2: Yes

2. Has the statistical analysis been performed appropriately and rigorously? 

Reviewer #1: Yes

Reviewer #2: Yes

3. Have the authors made all data underlying the findings in their manuscript fully available?

Reviewer #1: Yes

Reviewer #2: No

4. Is the manuscript presented in an intelligible fashion and written in standard English?

Reviewer #1: Yes

Reviewer #2: Yes

5. Review Comments to the Author

Reviewer #1: Important note: This review pertains only to ‘statistical aspects’ of the study and so ‘clinical aspects’ [like medical importance, relevance of the study, ‘clinical significance and implication(s)’ of the whole study, etc.] are to be evaluated [should be assessed] separately/independently. Further please note that any ‘statistical review’ is generally done under the assumption that (such) study specific methodological [as well as execution] issues are perfectly taken care of by the investigator(s). This review is not an exception to that and so does not cover clinical aspects {however, seldom comments are made only if those issues are intimately / scientifically related & intermingle with ‘statistical aspects’ of the study}. Agreed that ‘statistical methods’ are used as just tools here, however, they are vital part of methodology [and so should be given due importance].

COMMENTS: It is definitely a good study and is planned as well as executed nicely. However, I have few doubts [these respected actions may be correct but need explanations/clarifications/justifications. Please take them as suggestions]. The first is: ‘Why there were two control groups [active control (‘HE,’ n=41), or waitlist control (‘WL,’ n=42)] in the study?’ [You may know that ‘Permuted Block Randomization’ ensures same group sizes (not]. Role/importance/necessity of ‘waitlist controls’ in any Psychology study is well-known. However, why two control groups?

Next is: Tool used to measure ‘perceived stress’ should have been mentioned in the abstract itself [as it is a Primary Outcome Measure]. In the ‘Abstract-Results’ section you said “Those with greater adherence to MED or MED+HE had greater reductions in stress [, cravings, and adiposity (ps<.05)]” but not mentioned tool or table (where this change is displayed and tested). Much later in line 156, you mentioned that the Perceived Stress Scale (PSS) was used. Is that alright?

The Food Acceptance and Awareness Questionnaire (FAAQ) was used to measure acceptance of urges and cravings to eat. Since the FAAQ is {made up of 10 items, each} rated on a 6-point Likert scale (1=very seldom true to 6=always true) and might have included ‘not true’ (negative) response also, which needs reverse scoring (very often). Also in this context, please note that the following {which is pasted from one standard textbook on ‘Research Methodology’ and I am sure that the authors already know these things, however, it is very essential to keep the limitations in mind while interpreting results [note that I am not asking you to change the study design]}:

Whenever response options ranged from 1=strongly disagree to 4=strongly agree (or ranging from 1 (strongly disagree) to 6 (strongly agree) or from 1=very bad to 3=neither good nor bad to 5=very good), while using a ‘Likert’ scale responses, recoding [like strongly disagree=-2, disagree=-1, neutral=0, agree=1, strongly agree=2] may yield correct and meaningful ‘arithmetic mean’ which is useful not only for comparison but has absolute meaning, in my opinion. Application of any statistical test(s) assume that meaning of entity used (mean, SD, etc) has a particular meaning. Though ‘α’ [alpha] or most other measures of reliability/correlation will remain same, however, use of non-parametric methods should/may be preferred while dealing with data yielded by any questionnaire/score.

Further note that though the measures/tools used are appropriate, most of them yield data that are in [at the most] ‘ordinal’ level of measurement [and not in ratio level of measurement for sure {as the score two times higher does not indicate presence of that parameter/phenomenon as double (for example, a Visual Analogue Scales VAS score or say ‘depression/stress’ score)}]. Then application of suitable non-parametric test(s) is/are indicated/advisable [even if distribution may be ‘Gaussian’ (i.e., normal)]. Agreed that there is/are no non-parametric test(s)/technique(s) available to be used as alternative in all situation(s) [suitable / most desired/applicable], but should be used whenever/wherever they are available.

As you know well that while reporting [findings from] ‘Clinical Trial’ one should follow CONSORT guidelines. Even important items {like How sample size was determined (Item 7a), Random Sequence generation (Item 8a), Allocation concealment (Item 9), Blinding (Item 11a)} of/in CONSORT checklist are not found [since your article type is ‘Clinical Trial’, you are supposed to cover these items in the report]. How you arrived at this sample size [with complete estimation procedure] must be described in details as ultimately you had to say (lines 371-2 that ‘our study was likely limited by a sample size that may have been too small to detect modest interaction effects’}. Fig 1. CONSORT Flow Diagram is alright but covers only about flow of cases/numbers.

There are only two tables in the manuscript – one on baseline demographics and the other on baseline heath characteristics – remaining vital information [mainly comparison statistics] are put/presented in either text or figures. But remember that (in my opinion) figures are complementary and not alternatives of/to tables. One good thing is that there is no statistical comparison of baseline characteristics [read the following]:

To provide a description of baseline characteristics is entirely reasonable (since it is clearly important in assessing to whom the results of the trial can be applied), however, statistical comparison of baseline characteristics is not desirable at all [because even if P-value turns out to be significant (while comparing baseline characteristics despite random allocation), it is, by definition, a false positive] as you then are supposed to be testing ‘randomization’ then, which in any single trial may not balance all baseline characteristics [particularly when sample sizes are small] because ‘randomization’ is a sort of ‘insurance’ and not a guarantee scheme.

Is not it essential to adjust P-value(s) even if ‘series of Analysis of Covariance (ANCOVA)’ are used/applied as it a sort of multiple testing (multiple comparisons) problem/issue?

Except these few points, this manuscript is alright and I have no hesitation to recommend acceptance after minor revision.

Reviewer #2: Title: The coma (,) in the randomized controlled trial shall be excluded.

Materials and Methods:

Under “Interventions”, please specify the frequency (eg. on a daily basis or at least N number of days per week of the 8-weeks intervention. Also, please include how the researchers have verified whether the participants had used the app in the given period.

For HE Group, when was the counseling conducted? Was it at the beginning of the intervention?

Please elaborate on the “digitally-based mindful eating program”. Include name of the app, duration of the mindful eating practice and how the usage per user was assured.

Did you check if the waitlist control group had already access to Headspace or other mindfulness apps like Calm? Have you also considered previous experience of (all) the participants with regard to mindfulness or other meditation practices?

Under “Measures”, please include the reliability and validity of the instruments used.

Discussion and Conclusion: Please separate Conclusion as a distinct section. Highlight the implications of the study at the end of Discussion, and also include delimitations.

6. PLOS authors have the option to publish the peer review history of their article (what does this mean?). If published, this will include your full peer review and any attached files.

Reviewer #1: **Yes: **Dr. Sanjeev Sarmukaddam

Reviewer #2: **Yes: **Allen Joshua George

---

## [Author Response · Author response to Decision Letter 0]

3 Sep 2022

Dear Dr. Tischer,

We are pleased to submit a revised version of our research article, entitled Impact of digital meditation on work stress and health outcomes among adults with overweight: A randomized controlled trial for consideration of publication in PLOS ONE. We are grateful for the reviewers’ thoughtful feedback and have included our detailed responses below. We have also highlighted changes throughout our manuscript using tracked changes.

Response to Editor and Reviewer Comments: 

Editor’s comments:

• Provision of CONSORT checklist- this is now included

• Include marked up copy of manuscript with tracked changes as well as unmarked version without tracked changes- these are now included

Reviewer #1: 

• This review pertains only to ‘statistical aspects’ of the study and so ‘clinical aspects’ [like medical importance, relevance of the study, ‘clinical significance and implication(s)’ of the whole study, etc.] are to be evaluated [should be assessed] separately/independently. Further please note that any ‘statistical review’ is generally done under the assumption that (such) study specific methodological [as well as execution] issues are perfectly taken care of by the investigator(s). This review is not an exception to that and so does not cover clinical aspects (however, seldom comments are made only if those issues are intimately / scientifically related & intermingle with ‘statistical aspects’ of the study). Agreed that ‘statistical methods’ are used as just tools here, however, they are vital part of methodology [and so should be given due importance]. 

o RESPONSE: Noted. We appreciate this reviewer’s keen insights with regard to statistical aspects of this study. We feel this reviewer’s feedback has greatly enhanced the quality and rigor of the manuscript. 

• It is definitely a good study and is planned as well as executed nicely. 

o RESPONSE: We thank this reviewer for your positive feedback regarding the manuscript’s methodology.

• Why there were two control groups [active control (‘HE,’ n=41), or waitlist control (‘WL,’ n=42)] in the study?’ [You may know that ‘Permuted Block Randomization’ ensures same group sizes (not]. Role/importance/necessity of ‘waitlist controls’ in any Psychology study is well-known. However, why two control groups? 

o RESPONSE: We thank this reviewer for this important question. We felt that the addition of an active control group (‘healthy eating’) which did not include a general mindfulness component, but which was matched for time and attention and other non-specific intervention effects, would allow us to test whether meditation could “out-perform” both a waitlist control (with no treatment or attention components whatsoever) and an active control (that featured some clinical attention and some general health guidelines around eating, generally thought to be a non-specific treatment component for improving eating behavior). The issue of adding an active control group to clinical trials has been receiving more and more attention in recent years, particularly in psychological intervention research, and is generally considered advantageous to a purely waitlist control condition (Kinser & Robins, 2013). We have added this clarification to lines 102-103:

We aimed to test whether digital meditation could out-perform an active control condition that was matched for time and attention and other non-specific intervention effects (Kinser & Robins, 2013).

• Tool used to measure ‘perceived stress’ should have been mentioned in the abstract itself [as it is a Primary Outcome Measure]. In the ‘Abstract-Results’ section you said “Those with greater adherence to MED or MED+HE had greater reductions in stress [, cravings, and adiposity (ps<.05)]” but not mentioned tool or table (where this change is displayed and tested). Much later in line 156, you mentioned that the Perceived Stress Scale (PSS) was used. Is that alright? 

o RESPONSE: We thank this reviewer for noting this oversight. We have added more specificity about primary and secondary outcomes measures to the abstract, on lines 30-31: 

Participants (n=145; M(SD) BMI: 30.8 (5.4) kg/m2) completed baseline and 8-week measures of stress (Perceived Stress Scale), cravings (Food Acceptance and Awareness Questionnaire) and adiposity (sagittal diameter and BMI).

• The Food Acceptance and Awareness Questionnaire (FAAQ) was used to measure acceptance of urges and cravings to eat. Since the FAAQ is (made up of 10 items, each) rated on a 6-point Likert scale (1=very seldom true to 6=always true) and might have included ‘not true’ (negative) response also, which needs reverse scoring (very often). Also in this context, please note that the following (which is pasted from one standard textbook on ‘Research Methodology’ and I am sure that the authors already know these things, however, it is very essential to keep the limitations in mind while interpreting results [note that I am not asking you to change the study design]: Whenever response options ranged from 1=strongly disagree to 4=strongly agree (or ranging from 1 (strongly disagree) to 6 (strongly agree) or from 1=very bad to 3=neither good nor bad to 5=very good), while using a ‘Likert’ scale responses, recoding [like strongly disagree=-2, disagree=-1, neutral=0, agree=1, strongly agree=2] may yield correct and meaningful ‘arithmetic mean’ which is useful not only for comparison but has absolute meaning, in my opinion. Application of any statistical test(s) assume that meaning of entity used (mean, SD, etc) has a particular meaning. Though ‘α’ [alpha] or most other measures of reliability/correlation will remain same, however, use of non- parametric methods should/may be preferred while dealing with data yielded by any questionnaire/score. 

o RESPONSE: We thank this reviewer for noting the limitation of the way in which the FAAQ is scored using a 6-point Likert scale, without the option for negative or not true responses. We have kept the scoring to be consistent with conventional scoring approaches for this measure (Juarascio, Forman, Timko, Butryn, & Goodwin, 2011), to ensure ease of comparison of FAAQ scores across manuscripts. However, we acknowledge this scoring limitation in our discussion section, on lines 467-470:

Further, our measures of dysregulated eating may not fully reflect non-homeostatic eating behavior (vs. a semi-structured interview measure of eating pathology), and the scoring metrics for the FAAQ (a 6-point Likert scale ranging from 1 to 6) without the option for including negative (e.g., -1, -2) response may not yield particularly meaningful arithmetic means. 

• Further note that though the measures/tools used are appropriate, most of them yield data that are in [at the most] ‘ordinal’ level of measurement [and not in ratio level of measurement for sure (as the score two times higher does not indicate presence of that parameter/phenomenon as double (for example, a Visual Analogue Scales VAS score or say ‘depression/stress’ score)]. Then application of suitable non-parametric test(s) is/are indicated/advisable [even if distribution may be ‘Gaussian’ (i.e., normal)]. Agreed that there is/are no non-parametric test(s)/technique(s) available to be used as alternative in all situation(s) [suitable / most desired/applicable], but should be used whenever/wherever they are available. 

o RESPONSE: We thank this reviewer for noting the limitation of survey measures that yield data in the ordinal level of measurement (e.g., PSS, FAAQ). Per this reviewer’s suggestion, we have run non-parametric tests (Kruskal-Wallis test, instead of ANOVA) for outcome measures using ordinal level measurement (PSS and FAAQ) where possible, and yielded nearly identical findings as with our parametric tests. Due to space limitations we have not included these findings in the manuscript, but we acknowledge the following on lines 322-323, and lines 342-343:

Findings were identical when using non-parametric tests (Kruskal-Wallis), given the ordinal nature of the PSS scoring.

Findings were identical when using non-parametric tests (Kruskal-Wallis), given the ordinal nature of the FAAQ scoring.

• As you know well that while reporting [findings from] ‘Clinical Trial’ one should follow CONSORT guidelines. Even important items (like How sample size was determined (Item 7a), Random Sequence generation (Item 8a), Allocation concealment (Item 9), Blinding (Item 11a)) of/in CONSORT checklist are not found [since your article type is ‘Clinical Trial’, you are supposed to cover these items in the report]. How you arrived at this sample size [with complete estimation procedure] must be described in details as ultimately you had to say (lines 371-2 that ‘our study was likely limited by a sample size that may have been too small to detect modest interaction effects’). Fig 1. CONSORT Flow Diagram is alright but covers only about flow of cases/numbers. 

o RESPONSE: We thank this reviewer for noting this oversight. Most of these details required by CONSORT guidelines are contained within our Supplemental CONSORT Study protocol, which has now been submitted to the editor. We have also added most of these details throughout the manuscript. For instance, on lines 133-136 and on lines 142-148, respectively, we have added the following:

We aimed to enroll up to 150 participants. Our prior study (Bostock, Crosswell, Prather, & Steptoe, 2019) detected effects in a sample of <250 participants. We therefore expected that our sample size of 150 would be well-powered to detect improvements in our self-report measures in response to our treatment intervention. 

Study personnel then randomly assigned participants to one of four possible conditions, using factorial assignment, on Qualtrics…. Study personnel were not able to access the file containing the sequence of assignments or to see the next condition in the sequence until the moment they randomized the participant. 

• There are only two tables in the manuscript – one on baseline demographics and the other on baseline heath characteristics – remaining vital information [mainly comparison statistics] are put/presented in either text or figures. But remember that (in my opinion) figures are complementary and not alternatives of/to tables. One good thing is that there is no statistical comparison of baseline characteristics [read the following]:

To provide a description of baseline characteristics is entirely reasonable (since it is clearly important in assessing to whom the results of the trial can be applied), however, statistical comparison of baseline characteristics is not desirable at all [because even if P-value turns out to be significant (while comparing baseline characteristics despite random allocation), it is, by definition, a false positive] as you then are supposed to be testing ‘randomization’ then, which in any single trial may not balance all baseline characteristics [particularly when sample sizes are small] because ‘randomization’ is a sort of ‘insurance’ and not a guarantee scheme. 

o RESPONSE: We thank this reviewer for this comment and completely agree that baseline characteristics should not include any sort of statistical comparison. We also agree with this reviewer that figures are considered complementary, and have thus included all relevant and necessary statistics pertaining to these figures within the text of the manuscript. 

• Is not it essential to adjust P-value(s) even if ‘series of Analysis of Covariance (ANCOVA)’ are used/applied as it a sort of multiple testing (multiple comparisons) problem/issue? 

o RESPONSE: We thank this reviewer for this keen observation regarding multiple comparisons using ANCOVA. Notably, all of our results remained similar when adjusting for multiple comparisons (using a Bonferroni adjusted alpha level of .03 (.05/2) for each ANCOVA model). 

• Except these few points, this manuscript is alright and I have no hesitation to recommend acceptance after minor revision. 

o RESPONSE: We thank this reviewer for your positive feedback regarding the overall quality of the manuscript, and appreciate the keen attention to statistical concerns throughout. 

Reviewer #2:

• Title: The coma (,) in the randomized controlled trial shall be excluded. 

o RESPONSE: We thank this reviewer for noting this oversight and have removed the comma from the title (Line 4). 

• Materials and Methods: 

• Under “Interventions”, please specify the frequency (e.g.. on a daily basis or at least N number of days per week of the 8-weeks intervention. 

o RESPONSE: We have added frequency of contact/involvement for each intervention category. Lines 155-186 now say:

Meditation group (‘MED’)…Participants were expected to meditate 5 days per week over the course of 8 weeks

Healthy eating group (‘HE’)… Participants had a total of approximately 1.5-2 hours of contact with a counselor, and were expected to engage with the online resources 1 day per week over the course of 8 weeks.

Meditation + Healthy eating group (‘MED+HE’)… Participants were expected to meditate 5 days per week over the course of 8 weeks and they had a total of approximately 1.5-2 hours of contact with a counselor, and were expected to engage with the online resources 1 day per week over the course of 8 weeks.

Waitlist control condition (‘WL’)… Participants had no contact with a study counselor over the course of the 8 week intervention period.

• Also, please include how the researchers have verified whether the participants had used the app in the given period.

o RESPONSE: The research team had access to user data through Headspace, and were able to calculate total number of minutes spent meditating. This information has been added to lines 238-239:

The research team had access to individual user data via Headspace, in order to make these calculations.

• For HE Group, when was the counseling conducted? Was it at the beginning of the intervention?

o RESPONSE: The counseling session was conducted at the very beginning of the intervention, within week one. This information has been added to line 158. 

• Please elaborate on the “digitally-based mindful eating program.” Include name of the app, duration of the mindful eating practice and how the usage per user was assured.

o RESPONSE: The digitally-based mindful eating program was created specifically for this study by the research team, and was primarily a secured website that included information on mindful eating, and audio tools for mindful eating practice (~3-5 minute practices). These details have now been added to lines 164-166.

• Did you check if the waitlist control group had already access to Headspace or other mindfulness apps like Calm? Have you also considered previous experience of (all) the participants with regard to mindfulness or other meditation practices?

o RESPONSE: We did not provide Headspace access codes to WL participants until after they completed a 2-month follow-up questionnaire. However, we had no way of objectively verifying whether participants already had subscriptions to other mindfulness apps. We excluded individuals who indicated they were experienced meditators (defined as 3 times per week for 10 minutes or more), and at baseline, participants reported meditating less than once a week; in fact, the majority (79%) indicated that they had never meditated, and only 4% indicated that they meditated 1-2 times per week prior to treatment randomization (Table 2). Given that so few individuals indicated prior experience with meditation, we did not examine this baseline characteristic as a treatment covariate. We have noted the limitation of being unable to objectively verify access to meditation programs for those in our control conditions to lines 465-467:

We were unable to truly ascertain whether participants in either control condition were accessing mindfulness programs or apps during the 8 week intervention period. 

• Under “Measures”, please include the reliability and validity of the instruments used. 

o RESPONSE: We appreciate this feedback, and we have now included relevant psychometric properties of each of the instruments used (PSS, FAAQ, and QEWP-5) throughout the measures section (Lines 193-231). 

• Discussion and Conclusion: 

• Please separate Conclusion as a distinct section. 

o RESPONSE: We have now created a Conclusion section that is distinct from the Discussion section (Line 473), and which highlights the clinical implications of the study.

---

## [Decision Letter · Decision Letter 1]

4 Oct 2022

PONE-D-21-38719R1Impact of digital meditation on work stress and health outcomes among adults with overweight: A randomized controlled trialPLOS ONE

Dear Dr. Radin,

Thank you for submitting your manuscript to PLOS ONE. After careful consideration, we feel that it has merit but does not fully meet PLOS ONE’s publication criteria as it currently stands. Therefore, we invite you to submit a revised version of the manuscript that addresses the points raised during the review process. Thank you for revising the manuscript and providing detailed responses to the previous reviews. The two original reviewers provided positive comments on the revisions, and their comments are available below. The manuscript mentions a "digitally-based mindful eating program" which was created for this study and provided to participants via a secured website> However it is not clear what this program entailed, how it was designed and if it is intended to be provided as a commercial product. Please respond to the editor's queries regarding this issue. Editor's queries:Please provide further information on the “digitally-based mindful eating program.” 

- Please include in the manuscript a brief outline of the material in the digitally-based mindful eating program and how it was designed.

- Are there any previous publications describing the program or research that underpins it? If so please cite them in the manuscript.

- Is there any public-facing information about the program accessible to readers? If so, please add links to relevant webpages as citation(s). Guidelines for formatting references to online sources are here: https://journals.plos.org/plosone/s/submission-guidelines#loc-references

- Is the program associated with a commercial provider and/or patent or is it intended to be provided commercially in the future? If so, are any of the authors associated with the commercial provider or patent? Please review PLOS's Competing Interest policy and ensure all potential competing interests are declared (https://journals.plos.org/plosone/s/competing-interests) 

We look forward to receiving your revised manuscript.

Kind regards,

Clare Mc Fadden, PhD

Staff Editor

PLOS ONE

Journal Requirements:

Reviewers' comments:

Reviewer's Responses to Questions

**Comments to the Author**

1. If the authors have adequately addressed your comments raised in a previous round of review and you feel that this manuscript is now acceptable for publication, you may indicate that here to bypass the “Comments to the Author” section, enter your conflict of interest statement in the “Confidential to Editor” section, and submit your "Accept" recommendation.

Reviewer #1: All comments have been addressed

Reviewer #2: All comments have been addressed

2. Is the manuscript technically sound, and do the data support the conclusions?

Reviewer #1: (No Response)

Reviewer #2: Yes

3. Has the statistical analysis been performed appropriately and rigorously? 

Reviewer #1: (No Response)

Reviewer #2: Yes

4. Have the authors made all data underlying the findings in their manuscript fully available?

Reviewer #1: (No Response)

Reviewer #2: No

5. Is the manuscript presented in an intelligible fashion and written in standard English?

Reviewer #1: (No Response)

Reviewer #2: Yes

6. Review Comments to the Author

Reviewer #1: COMMENTS: Since all of the comments made on earlier draft are considered positively & attended, I recommend the acceptance. The manuscript now has achieved acceptable level, in my opinion.

Very nice that [“we have run non-parametric tests (Kruskal-Wallis test, instead of ANOVA) for outcome measures using ordinal level measurement (PSS and FAAQ) where possible”. However, remember that even if they yielded nearly identical findings as with parametric tests, it is always good to apply correct/indicated ones.

Reviewer #2: After revision the manuscript has better clarity and structure. The authors have addressed all the previous recommendations.

7. PLOS authors have the option to publish the peer review history of their article (what does this mean?). If published, this will include your full peer review and any attached files.

Reviewer #1: **Yes: **Dr. Sanjeev Sarmukaddam

Reviewer #2: **Yes: **Allen Joshua George

---

## [Author Response · Author response to Decision Letter 1]

17 Nov 2022

We are pleased to submit a revised version of our research article, entitled Impact of digital meditation on work stress and health outcomes among adults with overweight: A randomized controlled trial for consideration of publication in PLOS ONE. We are pleased to learn that the previous peer reviewers were satisfied with our edits and responsiveness to earlier feedback. We are grateful for your thoughtful additional feedback and have included our detailed responses below. We have also highlighted changes throughout our manuscript using tracked changes.

Response to Editor Comments: 

Editor’s queries:

• The manuscript mentions a "digitally-based mindful eating program" which was created for this study and provided to participants via a secured website. However, it is not clear what this program entailed, how it was designed and if it is intended to be provided as a commercial product. Please respond to the editor's queries regarding this issue. Please provide further information on the “digitally-based mindful eating program.” Please include in the manuscript a brief outline of the material in the digitally-based mindful eating program and how it was designed.

o RESPONSE: We developed a brief mindful eating program for this study, geared towards helping participants develop goals to improve eating behavior (e.g., to reduce compulsive eating and increase mindful eating). This program was in the early, pilot stages (NCT03945214, funded through the first author’s K23: 1K23AT011048) to assess feasibility and acceptability. Given its early stages of development, the program is not intended to be provided as a commercial product. It is in need of further refinement and tailoring for specific populations who struggle with binge eating before it can be offered as a commercial product. The program included a combination of both an in-person motivational-interviewing-based counseling session, text message support, and access to a website that provided audio exercises and tools to practice eating mindfully, including tools to ride out urges to overeat. The audio exercises were scripted and recorded by the study’s first author and adapted from a number of mindful eating resources including the MB-EAT curriculum. 

We developed the brief mindful eating program for this study, geared towards helping participants develop goals to reduce compulsive eating and increase mindful eating. This program was in the pilot stages to assess feasibility and acceptability. The program is adapted from several sources, including motivational interviewing for binge eating,1 weight management,2,3 and sugar-sweetened beverage intake (from our recently completed trial),4 and mindfulness-based eating awareness training.5 The program is comprised of an initial one-on-one counseling session, three booster calls during the 8-week intervention period, engagement with an online mindful eating program with instruction on mindful eating practices, and text message reminders of support three times a week. During the session, trained health counselors follow a semi-structured protocol of: (1) engaging participants in a conversation about the concerns they have about their eating (2) identifying vulnerabilities for challenging eating patterns, (3) asking participants to consider why/how making changes to eating would be important for them, (4) identifying perceived barriers to changing eating habits, (5) providing psychoeducation on sugar intake, stress eating, and their links with health, (6) collaborating with participants on specific, achievable eating-related goals, (7) identifying motivational factors, (8) engaging participants to rate level of confidence and self-efficacy in making changes, and (9) guiding participants through a brief mindful eating exercise using a bite-sized snack of their choice. Participants were then given access to a password-secured website which contained up to six different brief audio exercises using mindful eating and urge-surfing strategies. They were instructed to access these audios during high vulnerability times for compulsive eating. For example, for those who identify cravings as a potent trigger for problematic consumption, participants could access a brief urge-surfing exercise to learn how to ‘ride out’ a craving. 

The following details about the digital component of the program have now been added to the manuscript (lines 157-163): 

This password-secured website contained up to six different brief audio exercises using mindful eating and urge-surfing strategies. The audio exercises were scripted and recorded by the study’s first author and adapted from a number of mindful eating resources including the MB-EAT curriculum. We instructed participants to access these audios during high vulnerability times for compulsive eating. For example, for those who identify cravings as a potent trigger for problematic consumption, participants could access a brief urge-surfing exercise to learn how to ‘ride out’ a craving.

• Are there any previous publications describing the program or research that underpins it? If so please cite them in the manuscript.

o RESPONSE: The program is adapted from several sources, including motivational interviewing for binge eating,1 weight management,2,3 and sugar-sweetened beverage intake (from our recently completed trial),4 and mindfulness-based eating awareness training.5 However, the current manuscript is the first to describe the program in its current form. The following details have now been added to the manuscript (lines 165-167): 

The program is adapted from several sources, including motivational interviewing for binge eating,1 weight management,2,3 and sugar-sweetened beverage intake (from our recently completed trial),4 and mindfulness-based eating awareness training.5

• Is there any public-facing information about the program accessible to readers? If so, please add links to relevant webpages as citation(s). Guidelines for formatting references to online sources are here: hfps://journals.plos.org/plosone/s/submission-guidelines#loc-references

o RESPONSE: This program was in the early, pilot stages (NCT03945214, funded through the first author’s K23: 1K23AT011048) to assess feasibility and acceptability. Given its early stages of development, the program is not yet accessible to readers in public-facing formats such as public webpages. 

• Is the program associated with a commercial provider and/or patent or is it intended to be provided commercially in the future? If so, are any of the authors associated with the commercial provider or patent? Please review PLOS's Competing Interest policy and ensure all potential competing interests are declared (hfps://journals.plos.org/plosone/s/compeUng-interests) 

o RESPONSE: As described previously, this program was in the early, pilot stages (NCT03945214, funded through the first author’s K23: 1K23AT011048) to assess feasibility and acceptability. Given its early stages of development, the program is not intended to be provided as a commercial product. It is in need of further refinement and tailoring for specific populations who struggle with binge eating before it can be offered as a commercial product.

---

## [Editor Report · Decision Letter 2]

10 Jan 2023

Impact of digital meditation on work stress and health outcomes among adults with overweight: A randomized controlled trial

PONE-D-21-38719R2

Dear Dr. Radin,

We’re pleased to inform you that your manuscript has been judged scientifically suitable for publication and will be formally accepted for publication once it meets all outstanding technical requirements.

Kind regards,

Yann Benetreau

Staff Editor

PLOS ONE

Additional Editor Comments (optional):

With apologies for the lengthy review time. Please ensure that the formatting of references adheres to our submission guidelines at https://journals.plos.org/plosone/s/submission-guidelines#loc-references
---

## [Editor Report · Acceptance letter]

2 Feb 2023

PONE-D-21-38719R2 

Impact of digital meditation on work stress and health outcomes among adults with overweight: A randomized controlled trial 

Dear Dr. Radin:

I'm pleased to inform you that your manuscript has been deemed suitable for publication in PLOS ONE. Congratulations! Your manuscript is now with our production department. 

Kind regards, 

on behalf of

Dr. Yann Benetreau 

Staff Editor

PLOS ONE